# Multivariate Regression with Calibration[*]

**Han Liu**
Department of Operations Research and Financial Engineering
Princeton University

**Lie Wang**
Department of Mathematics
Massachusetts Institute of Technology

**Tuo Zhao**[†]
Department of Computer Science
Johns Hopkins University

## Abstract

We propose a new method named calibrated multivariate regression (CMR) for fitting high dimensional multivariate regression models. Compared to existing methods, CMR calibrates the regularization for each regression task with respect to its noise level so that it is simultaneously tuning insensitive and achieves an improved finite-sample performance. Computationally, we develop an efficient smoothed proximal gradient algorithm which has a worst-case iteration complexity $O(1/\epsilon)$, where $\epsilon$ is a pre-specified numerical accuracy. Theoretically, we prove that CMR achieves the optimal rate of convergence in parameter estimation. We illustrate the usefulness of CMR by thorough numerical simulations and show that CMR consistently outperforms other high dimensional multivariate regression methods. We also apply CMR on a brain activity prediction problem and find that CMR is as competitive as the handcrafted model created by human experts.

## 1 Introduction

Given a design matrix $\mathbf{X} \in \mathbb{R}^{n \times d}$ and a response matrix $\mathbf{Y} \in \mathbb{R}^{n \times m}$, we consider a multivariate linear model $\mathbf{Y} = \mathbf{X}\mathbf{B}^0 + \mathbf{Z}$, where $\mathbf{B}^0 \in \mathbb{R}^{d \times m}$ is an unknown regression coefficient matrix and $\mathbf{Z} \in \mathbb{R}^{n \times m}$ is a noise matrix [1]. For a matrix $\mathbf{A} = [\mathbf{A}_{jk}] \in \mathbb{R}^{d \times m}$, we denote $\mathbf{A}_{j*} = (\mathbf{A}_{j1}, ..., \mathbf{A}_{jm}) \in \mathbb{R}^m$ and $\mathbf{A}_{*k} = (\mathbf{A}_{1k}, ..., \mathbf{A}_{dk})^T \in \mathbb{R}^d$ to be its $j^{\text{th}}$ row and $k^{\text{th}}$ column respectively. We assume that all $\mathbf{Z}_{i*}$'s are independently sampled from an $m$-dimensional Gaussian distribution with mean $\mathbf{0}$ and covariance matrix $\mathbf{\Sigma} \in \mathbb{R}^{m \times m}$.

We can represent the multivariate linear model as an ensemble of univariate linear regression models: $\mathbf{Y}_{*k} = \mathbf{X}\mathbf{B}^0_{*k} + \mathbf{Z}_{*k}$, $k = 1, ..., m$. Then we get a multi-task learning problem [3, 2, 26]. Multi-task learning exploits shared common structure across tasks to obtain improved estimation performance. In the past decade, significant progress has been made towards designing a variety of modeling assumptions for multivariate regression.

A popular assumption is that all the regression tasks share a common sparsity pattern, i.e., many $\mathbf{B}^0_{j*}$'s are zero vectors. Such a joint sparsity assumption is a natural extension of that for univariate linear regressions. Similar to the $L_1$-regularization used in Lasso [23], we can adopt group regularization to obtain a good estimator of $\mathbf{B}^0$ [25, 24, 19, 13]. Besides the aforementioned approaches, there are other methods that aim to exploit the covariance structure of the noise matrix $\mathbf{Z}$ [7, 22]. For

[*]The authors are listed in alphabetical order. This work is partially supported by the grants NSF IIS1408910, NSF IIS1332109, NSF Grant DMS-1005539, NIH R01MH102339, NIH R01GM083084, and NIH R01HG06841.

[†]Tuo Zhao is also affiliated with Department of Operations Research and Financial Engineering at Princeton University.

instance, [22] assume that all $\mathbf{Z}_{i*}$'s follow a multivariate Gaussian distribution with a sparse inverse covariance matrix $\boldsymbol{\Omega} = \boldsymbol{\Sigma}^{-1}$. They propose an iterative algorithm to estimate sparse $\mathbf{B}^0$ and $\boldsymbol{\Omega}$ by maximizing the penalized Gaussian log-likelihood. Such an iterative procedure is effective in many applications, but the theoretical analysis is difficult due to its nonconvex formulation.

In this paper, we assume an uncorrelated structure for the noise matrix $\mathbf{Z}$, i.e., $\boldsymbol{\Sigma} = \text{diag}(\sigma_1^2, \sigma_2^2, \ldots, \sigma_{m-1}^2, \sigma_m^2)$. Under this setting, we can efficiently solve the resulting estimation problem with a convex program as follows

$$\widehat{\mathbf{B}} = \underset{\mathbf{B}}{\operatorname{argmin}} \frac{1}{\sqrt{n}} ||\mathbf{Y} - \mathbf{XB}||_{\mathrm{F}}^2 + \lambda ||\mathbf{B}||_{1,p}, \tag{1.1}$$

where $\lambda > 0$ is a tuning parameter, and $||\mathbf{A}||_{\mathrm{F}} = \sqrt{\sum_{j,k} \mathbf{A}_{jk}^2}$ is the Frobenius norm of a matrix $\mathbf{A}$. Popular choices of $p$ include $p = 2$ and $p = \infty$: $||\mathbf{B}||_{1,2} = \sum_{j=1}^d \sqrt{\sum_{k=1}^m \mathbf{B}_{jk}^2}$ and $||\mathbf{B}||_{1,\infty} = \sum_{j=1}^d \max_{1 \le k \le m} |\mathbf{B}_{jk}|$. Computationally, the optimization problem in (1.1) can be efficiently solved by some first order algorithms [11, 12, 4].

The problem with the uncorrelated noise structure is amenable to statistical analysis. Under suitable conditions on the noise and design matrices, let $\sigma_{\max} = \max_k \sigma_k$, if we choose $\lambda = 2c \cdot \sigma_{\max} \left( \sqrt{\log d} + m^{1-1/p} \right)$, for some $c > 1$, then the estimator $\widehat{\mathbf{B}}$ in (1.1) achieves the optimal rates of convergence[1] [13], i.e., there exists some universal constant $C$ such that with high probability, we have

$$\frac{1}{\sqrt{m}} ||\widehat{\mathbf{B}} - \mathbf{B}^0||_{\mathrm{F}} \le C \cdot \sigma_{\max} \left( \sqrt{\frac{s \log d}{nm}} + \sqrt{\frac{sm^{1-2/p}}{n}} \right),$$

where $s$ is the number of rows with non-zero entries in $\mathbf{B}^0$. However, the estimator in (1.1) has two drawbacks: (1) All the tasks are regularized by the same tuning parameter $\lambda$, even though different tasks may have different $\sigma_k$'s. Thus more estimation bias is introduced to the tasks with smaller $\sigma_k$'s to compensate the tasks with larger $\sigma_k$'s. In another word, these tasks are not calibrated. (2) The tuning parameter selection involves the unknown parameter $\sigma_{\max}$. This requires tuning the regularization parameter over a wide range of potential values to get a good finite-sample performance.

To overcome the above two drawbacks, we formulate a new convex program named calibrated multivariate regression (CMR). The CMR estimator is defined to be the solution of the following convex program:

$$\widehat{\mathbf{B}} = \underset{\mathbf{B}}{\operatorname{argmin}} ||\mathbf{Y} - \mathbf{XB}||_{2,1} + \lambda ||\mathbf{B}||_{1,p}, \tag{1.2}$$

where $||\mathbf{A}||_{2,1} = \sum_k \sqrt{\sum_j \mathbf{A}_{jk}^2}$ is the nonsmooth $L_{2,1}$ norm of a matrix $\mathbf{A} = [\mathbf{A}_{jk}] \in \mathbb{R}^{d \times m}$. This is a multivariate extension of the square-root Lasso [5]. Similar to the square-root Lasso, the tuning parameter selection of CMR does not involve $\sigma_{\max}$. Moreover, the $L_{2,1}$ loss function can be viewed as a special example of the weighted least square loss, which calibrates each regression task (See more details in §2). Thus CMR adapts to different $\sigma_k$'s and achieves better finite-sample performance than the ordinary multivariate regression estimator (OMR) defined in (1.1).

Since both the loss and penalty functions in (1.2) are nonsmooth, CMR is computationally more challenging than OMR. To efficiently solve CMR, we propose a smoothed proximal gradient (SPG) algorithm with an iteration complexity $O(1/\epsilon)$, where $\epsilon$ is the pre-specified accuracy of the objective value [18, 4]. Theoretically, we provide sufficient conditions under which CMR achieves the optimal rates of convergence in parameter estimation. Numerical experiments on both synthetic and real data show that CMR universally outperforms existing multivariate regression methods. For a brain activity prediction task, prediction based on the features selected by CMR significantly outperforms that based on the features selected by OMR, and is even competitive with that based on the handcrafted features selected by human experts.

**Notations:** Given a vector $\boldsymbol{v} = (v_1, \ldots, v_d)^T \in \mathbb{R}^d$, for $1 \le p \le \infty$, we define the $L_p$-vector norm of $\boldsymbol{v}$ as $||\boldsymbol{v}||_p = \left( \sum_{j=1}^d |v_j|^p \right)^{1/p}$ if $1 \le p < \infty$ and $||\boldsymbol{v}||_p = \max_{1 \le j \le d} |v_j|$ if $p = \infty$.

Given two matrices $\mathbf{A} = [\mathbf{A}_{jk}]$ and $\mathbf{C} = [\mathbf{C}_{jk}] \in \mathbb{R}^{d \times m}$, we define the inner product of $\mathbf{A}$ and $\mathbf{C}$ as $\langle \mathbf{A}, \mathbf{C} \rangle = \sum_{j=1}^{d} \sum_{k=1}^{m} \mathbf{A}_{jk} \mathbf{C}_{jk} = \mathrm{tr}(\mathbf{A}^T \mathbf{C})$, where $\mathrm{tr}(\mathbf{A})$ is the trace of a matrix $\mathbf{A}$. We use $\mathbf{A}_{*k} = (\mathbf{A}_{1k}, ..., \mathbf{A}_{dk})^T$ and $\mathbf{A}_{j*} = (\mathbf{A}_{j1}, ..., \mathbf{A}_{jm})$ to denote the $k^{\text{th}}$ column and $j^{\text{th}}$ row of $\mathbf{A}$. Let $\mathcal{S}$ be some subspace of $\mathbb{R}^{d \times m}$, we use $\mathbf{A}_{\mathcal{S}}$ to denote the projection of $\mathbf{A}$ onto $\mathcal{S}$: $\mathbf{A}_{\mathcal{S}} = \mathrm{argmin}_{\mathbf{C} \in \mathcal{S}} ||\mathbf{C} - \mathbf{A}||_{\mathrm{F}}^2$. Moreover, we define the Frobenius and spectral norms of $\mathbf{A}$ as $||\mathbf{A}||_{\mathrm{F}} = \sqrt{\langle \mathbf{A}, \mathbf{A} \rangle}$ and $||\mathbf{A}||_2 = \psi_1(\mathbf{A})$, $\psi_1(\mathbf{A})$ is the largest singular value of $\mathbf{A}$. In addition, we define the matrix block norms as $||\mathbf{A}||_{2,1} = \sum_{k=1}^{m} ||\mathbf{A}_{*k}||_2$, $||\mathbf{A}||_{2,\infty} = \max_{1 \leq k \leq m} ||\mathbf{A}_{*k}||_2$, $||\mathbf{A}||_{1,p} = \sum_{j=1}^{d} ||\mathbf{A}_{j*}||_p$, and $||\mathbf{A}||_{\infty,q} = \max_{1 \leq j \leq d} ||\mathbf{A}_{j*}||_q$, where $1 \leq p \leq \infty$ and $1 \leq q \leq \infty$. It is easy to verify that $||\mathbf{A}||_{2,1}$ is the dual norm of $||\mathbf{A}||_{2,\infty}$. Let $1/\infty = 0$, then if $1/p + 1/q = 1$, $||\mathbf{A}||_{\infty,q}$ and $||\mathbf{A}||_{1,p}$ are also dual norms of each other.

## 2  Method

We solve the multivariate regression problem by the following convex program,

$$\widehat{\mathbf{B}} = \underset{\mathbf{B}}{\mathrm{argmin}} \, ||\mathbf{Y} - \mathbf{X}\mathbf{B}||_{2,1} + \lambda ||\mathbf{B}||_{1,p}. \tag{2.1}$$

The only difference between (2.1) and (1.1) is that we replace the $L_2$-loss function by the nonsmooth $L_{2,1}$-loss function. The $L_{2,1}$-loss function can be viewed as a special example of the weighted square loss function. More specifically, we consider the following optimization problem,

$$\widehat{\mathbf{B}} = \underset{\mathbf{B}}{\mathrm{argmin}} \sum_{k=1}^{m} \frac{1}{\sigma_k \sqrt{n}} ||\mathbf{Y}_{*k} - \mathbf{X}\mathbf{B}_{*k}||_2^2 + \lambda ||\mathbf{B}||_{1,p}, \tag{2.2}$$

where $\frac{1}{\sigma_k \sqrt{n}}$ is a weight assigned to calibrate the $k^{\text{th}}$ regression task. Without prior knowledge on $\sigma_k$'s, we use the following replacement of $\sigma_k$'s,

$$\widetilde{\sigma}_k = \frac{1}{\sqrt{n}} ||\mathbf{Y}_{*k} - \mathbf{X}\mathbf{B}_{*k}||_2, \ k = 1, ..., m. \tag{2.3}$$

By plugging (2.3) into the objective function in (2.2), we get (2.1). In another word, CMR calibrates different tasks by solving a penalized weighted least square program with weights defined in (2.3).

The optimization problem in (2.1) can be solved by the alternating direction method of multipliers (ADMM) with a global convergence guarantee [20]. However, ADMM does not take full advantage of the problem structure in (2.1). For example, even though the $L_{2,1}$ norm is nonsmooth, it is nondifferentiable only when a task achieves exact zero residual, which is unlikely in applications. In this paper, we apply the dual smoothing technique proposed by [18] to obtain a smooth surrogate function so that we can avoid directly evaluating the subgradient of the $L_{2,1}$ loss function. Thus we gain computational efficiency like other smooth loss functions.

We consider the Fenchel's dual representation of the $L_{2,1}$ loss:

$$||\mathbf{Y} - \mathbf{X}\mathbf{B}||_{2,1} = \max_{||\mathbf{U}||_{2,\infty} \leq 1} \langle \mathbf{U}, \mathbf{Y} - \mathbf{X}\mathbf{B} \rangle. \tag{2.4}$$

Let $\mu > 0$ be a smoothing parameter. The smooth approximation of the $L_{2,1}$ loss can be obtained by solving the following optimization problem

$$||\mathbf{Y} - \mathbf{X}\mathbf{B}||_{\mu} = \max_{||\mathbf{U}||_{2,\infty} \leq 1} \langle \mathbf{U}, \mathbf{Y} - \mathbf{X}\mathbf{B} \rangle - \frac{\mu}{2} ||\mathbf{U}||_{\mathrm{F}}^2, \tag{2.5}$$

where $||\mathbf{U}||_{\mathrm{F}}^2$ is the proximity function. Due to the fact that $||\mathbf{U}||_{\mathrm{F}}^2 \leq m ||\mathbf{U}||_{2,\infty}^2$, we obtain the following uniform bound by combing (2.4) and (2.5),

$$||\mathbf{Y} - \mathbf{X}\mathbf{B}||_{2,1} - \frac{m\mu}{2} \leq ||\mathbf{Y} - \mathbf{X}\mathbf{B}||_{\mu} \leq ||\mathbf{Y} - \mathbf{X}\mathbf{B}||_{2,1}. \tag{2.6}$$

From (2.6), we see that the approximation error introduced by the smoothing procedure can be controlled by a suitable $\mu$. Figure 2.1 shows several two-dimensional examples of the $L_2$ norm smoothed by different $\mu$'s. The optimization problem in (2.5) has a closed form solution $\widehat{\mathbf{U}}^{\mathbf{B}}$ with $\widehat{\mathbf{U}}_{*k}^{\mathbf{B}} = (\mathbf{Y}_{*k} - \mathbf{X}\mathbf{B}_{*k}) / \max \{||\mathbf{Y}_{*k} - \mathbf{X}\mathbf{B}_{*k}||_2, \mu\}$.

The next lemma shows that $||\mathbf{Y} - \mathbf{X}\mathbf{B}||_{\mu}$ is smooth in $\mathbf{B}$ with a simple form of gradient.

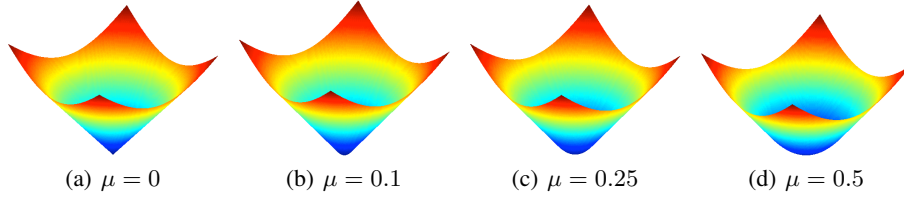

| (a) $\mu = 0$ | (b) $\mu = 0.1$ | (c) $\mu = 0.25$ | (d) $\mu = 0.5$ |

Figure 2.1: The $L_2$ norm ($\mu = 0$) and its smooth surrogates with $\mu = 0.1, 0.25, 0.5$. A larger $\mu$ makes the approximation more smooth, but introduces a larger approximation error.

**Lemma 2.1.** *For any $\mu > 0$, $||\mathbf{Y} - \mathbf{XB}||_\mu$ is a convex and continuously differentiable function in $\mathbf{B}$. In addition, $\mathbf{G}^\mu(\mathbf{B})$—the gradient of $||\mathbf{Y} - \mathbf{XB}||_\mu$ w.r.t. $\mathbf{B}$—has the form*

$$\mathbf{G}^\mu(\mathbf{B}) = \frac{\partial\left(\langle\widehat{\mathbf{U}}^{\mathbf{B}}, \mathbf{Y} - \mathbf{XB}\rangle + \mu||\widehat{\mathbf{U}}^{\mathbf{B}}||_{\mathrm{F}}^2/2\right)}{\partial\mathbf{B}} = -\mathbf{X}^T\widehat{\mathbf{U}}^{\mathbf{B}}. \tag{2.7}$$

*Moreover, let $\gamma = ||\mathbf{X}||_2^2$, then we have that $\mathbf{G}^\mu(\mathbf{B})$ is Lipschitz continuous in $\mathbf{B}$ with the Lipschitz constant $\gamma/\mu$, i.e., for any $\mathbf{B}', \mathbf{B}'' \in \mathbb{R}^{d\times m}$,*

$$||\mathbf{G}^\mu(\mathbf{B}') - \mathbf{G}^\mu(\mathbf{B}'')||_{\mathrm{F}} = ||\langle\mathbf{X}, \widehat{\mathbf{U}}^{\mathbf{B}'} - \widehat{\mathbf{U}}^{\mathbf{B}''}\rangle||_{\mathrm{F}} \le \frac{1}{\mu}||\mathbf{X}^T\mathbf{X}(\mathbf{B}' - \mathbf{B}'')||_{\mathrm{F}} \le \frac{\gamma}{\mu}||\mathbf{B}' - \mathbf{B}''||_{\mathrm{F}}.$$

Lemma 2.1 is a direct result of Theorem 1 in [18] and implies that $||\mathbf{Y} - \mathbf{XB}||_\mu$ has good computational structure. Therefore we apply the smooth proximal gradient algorithm to solve the smoothed version of the optimization problem as follows,

$$\widetilde{\mathbf{B}} = \underset{\mathbf{B}}{\operatorname{argmin}} ||\mathbf{Y} - \mathbf{XB}||_\mu + \lambda||\mathbf{B}||_{1,p}. \tag{2.8}$$

We then adopt the fast proximal gradient algorithm to solve (2.8) [4]. To derive the algorithm, we first define three sequences of auxiliary variables $\{\mathbf{A}^{(t)}\}$, $\{\mathbf{V}^{(t)}\}$, and $\{\mathbf{H}^{(t)}\}$ with $\mathbf{A}^{(0)} = \mathbf{H}^{(0)} = \mathbf{V}^{(0)} = \mathbf{B}^{(0)}$, a sequence of weights $\{\theta_t = 2/(t+1)\}$, and a nonincreasing sequence of step-sizes $\{\eta_t > 0\}$. For simplicity, we can set $\eta_t = \mu/\gamma$. In practice, we use the backtracking line search to dynamically adjust $\eta_t$ to boost the performance. At the $t^{\text{th}}$ iteration, we first take $\mathbf{V}^{(t)} = (1 - \theta_t)\mathbf{B}^{(t-1)} + \theta_t\mathbf{A}^{(t-1)}$. We then consider a quadratic approximation of $||\mathbf{Y} - \mathbf{XH}||_\mu$ as

$$Q\left(\mathbf{H}, \mathbf{V}^{(t)}, \eta_t\right) = ||\mathbf{Y} - \mathbf{XV}^{(t)}||_\mu + \langle\mathbf{G}^\mu(\mathbf{V}^{(t)}), \mathbf{H} - \mathbf{V}^{(t)}\rangle + \frac{1}{2\eta_t}||\mathbf{H} - \mathbf{V}^{(t)}||_{\mathrm{F}}^2.$$

Consequently, let $\widetilde{\mathbf{H}}^{(t)} = \mathbf{V}^{(t)} - \eta_t\mathbf{G}^\mu(\mathbf{V}^{(t)})$, we take

$$\mathbf{H}^{(t)} = \underset{\mathbf{H}}{\operatorname{argmin}} \, Q\left(\mathbf{H}, \mathbf{V}^{(t)}, \eta_t\right) + \lambda||\mathbf{H}||_{1,p} = \underset{\mathbf{H}}{\operatorname{argmin}} \, \frac{1}{2\eta_t}||\mathbf{H} - \widetilde{\mathbf{H}}^{(t)}||_{\mathrm{F}}^2 + \lambda||\mathbf{H}||_{1,p}. \tag{2.9}$$

When $p = 2$, (2.9) has a closed form solution $\mathbf{H}_{j*}^{(t)} = \widetilde{\mathbf{H}}_{j*} \cdot \max\left\{1 - \eta_t\lambda/||\widetilde{\mathbf{H}}_{j*}||_2, 0\right\}$. More details about other choices of $p$ in the $L_{1,p}$ norm can be found in [11] and [12]. To ensure that the objective value is nonincreasing, we choose

$$\mathbf{B}^{(t)} = \underset{\mathbf{B}\in\{\mathbf{H}^{(t)}, \mathbf{B}^{(t-1)}\}}{\operatorname{argmin}} ||\mathbf{Y} - \mathbf{XB}||_\mu + \lambda||\mathbf{B}||_{1,p}. \tag{2.10}$$

At last, we take $\mathbf{A}^{(t)} = \mathbf{B}^{(t-1)} + \frac{1}{\theta_t}(\mathbf{H}^{(t)} - \mathbf{B}^{(t-1)})$. The algorithm stops when $||\mathbf{H}^{(t)} - \mathbf{V}^{(t)}||_{\mathrm{F}} \le \varepsilon$, where $\varepsilon$ is the stopping precision.

The numerical rate of convergence of the proposed algorithm with respect to the original optimization problem (2.1) is presented in the following theorem.

**Theorem 2.2.** *Given a pre-specified accuracy $\epsilon$ and let $\mu = \epsilon/m$, after $t = 2\sqrt{m\gamma}||\mathbf{B}^{(0)} - \widehat{\mathbf{B}}||_{\mathrm{F}}/\epsilon - 1 = O(1/\epsilon)$ iterations, we have $||\mathbf{Y} - \mathbf{XB}^{(t)}||_{2,1} + \lambda||\mathbf{B}^{(t)}||_{1,p} \le ||\mathbf{Y} - \mathbf{X}\widehat{\mathbf{B}}||_{2,1} + \lambda||\widehat{\mathbf{B}}||_{1,p} + \epsilon$.*

The proof of Theorem 2.2 is provided in Appendix A.1. This result achieves the minimax optimal rate of convergence over all first order algorithms [18].

# 3 Statistical Properties

For notational simplicity, we define a re-scaled noise matrix $\mathbf{W} = [\mathbf{W}_{ik}] \in \mathbb{R}^{n \times m}$ with $\mathbf{W}_{ik} = \mathbf{Z}_{ik}/\sigma_k$, where $\mathbb{E}\mathbf{Z}_{ik}^2 = \sigma_k^2$. Thus $\mathbf{W}$ is a random matrix with all entries having mean 0 and variance 1. We define $\mathbf{G}^0$ to be the gradient of $||\mathbf{Y} - \mathbf{XB}||_{2,1}$ at $\mathbf{B} = \mathbf{B}^0$. It is easy to see that

$$\mathbf{G}_{*k}^0 = \frac{\mathbf{X}^T \mathbf{Z}_{*k}}{||\mathbf{Z}_{*k}||_2} = \frac{\mathbf{X}^T \mathbf{W}_{*k} \sigma_k}{||\mathbf{W}_{*k}\sigma_k||_2} = \frac{\mathbf{X}^T \mathbf{W}_{*k}}{||\mathbf{W}_{*k}||_2}$$

does not depend on the unknown quantities $\sigma_k$ for all $k = 1,...,m$. $\mathbf{G}_{*k}^0$ works as an important pivotal in our analysis. Moreover, our analysis exploits the decomposability of the $L_{1,p}$ norm [17]. More specifically, we assume that $\mathbf{B}^0$ has $s$ rows with all zero entries and define

$$\mathcal{S} = \left\{ \mathbf{C} \in \mathbb{R}^{d \times m} \mid \mathbf{C}_{j*} = \mathbf{0} \text{ for all } j \text{ such that } \mathbf{B}_{j*}^0 = \mathbf{0} \right\}, \tag{3.1}$$

$$\mathcal{N} = \left\{ \mathbf{C} \in \mathbb{R}^{d \times m} \mid \mathbf{C}_{j*} = \mathbf{0} \text{ for all } j \text{ such that } \mathbf{B}_{j*}^0 \neq \mathbf{0} \right\}. \tag{3.2}$$

Note that we have $\mathbf{B}^0 \in \mathcal{S}$ and the $L_{1,p}$ norm is decomposable with respect to the pair $(\mathcal{S}, \mathcal{N})$, i.e.,

$$||\mathbf{A}||_{1,p} = ||\mathbf{A}_{\mathcal{S}}||_{1,p} + ||\mathbf{A}_{\mathcal{N}}||_{1,p}.$$

The next lemma shows that when $\lambda$ is suitably chosen, the solution to the optimization problem in (2.1) lies in a restricted set.

**Lemma 3.1.** *Let* $\mathbf{B}^0 \in \mathcal{S}$ *and* $\widehat{\mathbf{B}}$ *be the optimum to (2.1), and* $1/p + 1/q = 1$. *We denote the estimation error as* $\widehat{\boldsymbol{\Delta}} = \widehat{\mathbf{B}} - \mathbf{B}^0$. *If* $\lambda \geq c||\mathbf{G}^0||_{\infty,q}$ *for some* $c > 1$, *we have*

$$\widehat{\boldsymbol{\Delta}} \in \mathcal{M}_c := \left\{ \boldsymbol{\Delta} \in \mathbb{R}^{d \times m} \mid ||\boldsymbol{\Delta}_{\mathcal{N}}||_{1,p} \leq \frac{c+1}{c-1}||\boldsymbol{\Delta}_{\mathcal{S}}||_{1,p} \right\}. \tag{3.3}$$

The proof of Lemma 3.1 is provided in Appendix B.1. To prove the main result, we also need to assume that the design matrix $\mathbf{X}$ satisfies the following condition.

**Assumption 3.1.** *Let* $\mathbf{B}^0 \in \mathcal{S}$, *then there exist positive constants* $\kappa$ *and* $c > 1$ *such that*

$$\kappa \leq \min_{\boldsymbol{\Delta} \in \mathcal{M}_c \setminus \{\mathbf{0}\}} \frac{||\mathbf{X}\boldsymbol{\Delta}||_{\mathrm{F}}}{\sqrt{n}||\boldsymbol{\Delta}||_{\mathrm{F}}}.$$

Assumption 3.1 is the generalization of the restricted eigenvalue conditions for analyzing univariate sparse linear models [17, 15, 6], Many common examples of random design satisfy this assumption [13, 21].

Note that Lemma 3.1 is a deterministic result of the CMR estimator for a fixed $\lambda$. Since $\mathbf{G}$ is essentially a random matrix, we need to show that $\lambda \geq cR^*(\mathbf{G}^0)$ holds with high probability to deliver a concrete rate of convergence for the CMR estimator in the next theorem.

**Theorem 3.2.** *We assume that each column of* $\mathbf{X}$ *is normalized as* $m^{1/2-1/p}||\mathbf{X}_{*j}||_2 = \sqrt{n}$ *for all* $j = 1,...,d$. *Then for some universal constant* $c_0$ *and large enough* $n$, *taking*

$$\lambda = \frac{2c(m^{1-1/p} + \sqrt{\log d})}{\sqrt{1-c_0}}, \tag{3.4}$$

*with probability at least* $1 - 2\exp(-2\log d) - 2\exp\left(-nc_0^2/8 + \log m\right)$, *we have*

$$\frac{1}{\sqrt{m}}||\widehat{\mathbf{B}} - \mathbf{B}^0||_{\mathrm{F}} \leq \frac{16c\sigma_{\max}}{\kappa^2(c-1)} \sqrt{\frac{1+c_0}{1-c_0}} \left( \sqrt{\frac{sm^{1-2/p}}{n}} + \sqrt{\frac{s\log d}{nm}} \right).$$

The proof of Theorem 3.2 is provided in Appendix B.2. Note that when we choose $p = 2$, the column normalization condition is reduced to $||\mathbf{X}_{*j}||_2 = \sqrt{n}$. Meanwhile, the corresponding error bound is further reduced to

$$\frac{1}{\sqrt{m}}||\widehat{\mathbf{B}} - \mathbf{B}^0||_{\mathrm{F}} = O_P\left( \sqrt{\frac{s}{n}} + \sqrt{\frac{s\log d}{nm}} \right),$$

which achieves the minimax optimal rate of convergence presented in [13]. See Theorem 6.1 in [13] for more technical details. From Theorem 3.2, we see that CMR achieves the same rates of convergence as the noncalibrated counterpart, but the tuning parameter $\lambda$ in (3.4) does not involve $\sigma_k$'s. Therefore CMR not only calibrates all the regression tasks, but also makes the tuning parameter selection insensitive to $\sigma_{\max}$.

# 4 Numerical Simulations

To compare the finite-sample performance between the calibrated multivariate regression (CMR) and ordinary multivariate regression (OMR), we generate a training dataset of 200 samples. More specifically, we use the following data generation scheme: (1) Generate each row of the design matrix $\mathbf{X}_{i*}$, $i = 1, ..., 200$, independently from a 800-dimensional normal distribution $N(\mathbf{0}, \boldsymbol{\Sigma})$ where $\boldsymbol{\Sigma}_{jj} = 1$ and $\boldsymbol{\Sigma}_{j\ell} = 0.5$ for all $\ell \neq j$.(2) Let $k = 1, \ldots, 13$, set the regression coefficient matrix $\mathbf{B}^0 \in \mathbb{R}^{800 \times 13}$ as $\mathbf{B}^0_{1k} = 3$, $\mathbf{B}^0_{2k} = 2$, $\mathbf{B}^0_{4k} = 1.5$, and $\mathbf{B}^0_{jk} = 0$ for all $j \neq 1, 2, 4$. (3) Generate the random noise matrix $\mathbf{Z} = \mathbf{WD}$, where $\mathbf{W} \in \mathbb{R}^{200 \times 13}$ with all entries of $\mathbf{W}$ are independently generated from $N(0, 1)$, and $\mathbf{D}$ is either of the following matrices

$$\mathbf{D}_I = \sigma_{\max} \cdot \mathrm{diag}\left(2^{0/4}, 2^{-1/4}, \cdots, 2^{-11/4}, 2^{-12/4}\right) \in \mathbb{R}^{13 \times 13}$$

$$\mathbf{D}_H = \sigma_{\max} \cdot \mathbf{I} \in \mathbb{R}^{13 \times 13}.$$

We generate a validation set of 200 samples for the regularization parameter selection and a testing set of 10,000 samples to evaluate the prediction accuracy.

In numerical experiments, we set $\sigma_{\max} = 1$, $2$, and $4$ to illustrate the tuning insensitivity of CMR. The regularization parameter $\lambda$ of both CMR and OMR is chosen over a grid $\boldsymbol{\Lambda} = \left\{2^{40/4}\lambda_0, 2^{39/4}\lambda_0, \cdots, 2^{-17/4}\lambda_0, 2^{-18/4}\lambda_0\right\}$, where $\lambda_0 = \sqrt{\log d} + \sqrt{m}$. The optimal regularization parameter $\widehat{\lambda}$ is determined by the prediction error as $\widehat{\lambda} = \mathrm{argmin}_{\lambda \in \boldsymbol{\Lambda}} ||\widetilde{\mathbf{Y}} - \widetilde{\mathbf{X}}\widehat{\mathbf{B}}^\lambda||_\mathrm{F}^2$, where $\widehat{\mathbf{B}}^\lambda$ denotes the obtained estimate using the regularization parameter $\lambda$, and $\widetilde{\mathbf{X}}$ and $\widetilde{\mathbf{Y}}$ denote the design and response matrices of the validation set.

Since the noise level $\sigma_k$'s are different in regression tasks, we adopt the following three criteria to evaluate the empirical performance: Pre. Err. $= \frac{1}{10000}||\overline{\mathbf{Y}} - \overline{\mathbf{X}}\widehat{\mathbf{B}}||_\mathrm{F}$, Adj. Pre. Err. $= \frac{1}{10000m}||(\overline{\mathbf{Y}} - \overline{\mathbf{X}}\widehat{\mathbf{B}})\mathbf{D}^{-1}||_\mathrm{F}^2$, and Est. Err. $= \frac{1}{m}||\widehat{\mathbf{B}} - \mathbf{B}^0||_\mathrm{F}^2$, where $\overline{\mathbf{X}}$ and $\overline{\mathbf{Y}}$ denotes the design and response matrices of the testing set.

All simulations are implemented by MATLAB using a PC with Intel Core i5 3.3GHz CPU and 16GB memory. CMR is solved by the proposed smoothing proximal gradient algorithm, where we set the stopping precision $\varepsilon = 10^{-4}$, the smoothing parameter $\mu = 10^{-4}$. OMR is solved by the monotone fast proximal gradient algorithm, where we set the stopping precision $\varepsilon = 10^{-4}$. We set $p = 2$, but the extension to arbitrary $p > 2$ is straightforward.

We first compare the smoothed proximal gradient (SPG) algorithm with the ADMM algorithm (the detailed derivation of ADMM can be found in Appendix A.2). We adopt the backtracking line search to accelerate both algorithms with a shrinkage parameter $\alpha = 0.8$. We set $\sigma_{\max} = 2$ for the adopted multivariate linear models. We conduct 200 simulations. The results are presented in Table 4.1. The SPG and ADMM algorithms attain similar objective values, but SPG is up to 4 times faster than ADMM. Both algorithms also achieve similar estimation errors.

We then compare the statistical performance between CMR and OMR. Tables 4.2 and 4.3 summarize the results averaged over 200 replicates. In addition, we also present the results of the oracle estimator, which is obtained by solving (2.2), since we know the true values of $\sigma_k$'s. Note that the oracle estimator is only for comparison purpose, and it is not a practical estimator. Since CMR calibrates the regularization for each task with respect to $\sigma_k$, CMR universally outperforms OMR, and achieves almost the same performance as the oracle estimator when we adopt the scale matrix $\mathbf{D}_I$ to generate the random noise. Meanwhile, when we adopt the scale matrix $\mathbf{D}_H$, where all $\sigma_k$'s are the same, CMR and OMR achieve similar performance. This further implies that CMR can be a safe replacement of OMR for multivariate regressions.

In addition, we also examine the optimal regularization parameters for CMR and OMR over all replicates. We visualize the distribution of all 200 selected $\widehat{\lambda}$'s using the kernel density estimator. In particular, we adopt the Gaussian kernel, and the kernel bandwidth is selected based on the 10-fold cross validation. Figure 4.1 illustrates the estimated density functions. The horizontal axis corresponds to the rescaled regularization parameter as $\log\left(\frac{\widehat{\lambda}}{\sqrt{\log d} + \sqrt{m}}\right)$. We see that the optimal regularization parameters of OMR significantly vary with different $\sigma_{\max}$. In contrast, the optimal regularization parameters of CMR are more concentrated. This is inconsistent with our claimed tuning insensitivity.

Table 4.1: Quantitive comparison of the computational performance between SPG and ADMM with the noise matrices generated using $\mathbf{D}_I$. The results are averaged over 200 replicates with standard errors in parentheses. SPG and ADMM attain similar objective values, but SPG is up to about 4 times faster than ADMM.

| $\lambda$ | Algorithm | Timing (second) | Obj. Val. | Num. Ite. | Est. Err. |
|---|---|---|---|---|---|
| $2\lambda_0$ | SPG | 2.8789(0.3141) | 508.21(3.8498) | 493.26(52.268) | 0.1213(0.0286) |
| | ADMM | 8.4731(0.8387) | 508.22(3.7059) | 437.7(37.4532) | 0.1215(0.0291) |
| $\lambda_0$ | SPG | 3.2633(0.3200) | 370.53(3.6144) | 565.80(54.919) | 0.0819(0.0205) |
| | ADMM | 11.976(1.460) | 370.53(3.4231) | 600.94(74.629) | 0.0822(0.0233) |
| $0.5\lambda_0$ | SPG | 3.7868(0.4551) | 297.24(3.6125) | 652.53(78.140) | 0.1399(0.0284) |
| | ADMM | 18.360(1.9678) | 297.25(3.3863) | 1134.0(136.08) | 0.1409(0.0317) |

Table 4.2: Quantitive comparison of the statistical performance between CMR and OMR with the noise matrices generated using $\mathbf{D}_I$. The results are averaged over 200 simulations with the standard errors in parentheses. CMR universally outperforms OMR, and achieves almost the same performance as the oracle estimator.

| $\sigma_{\max}$ | Method | Pre. Err. | Adj. Pre.Err | Est. Err. |
|---|---|---|---|---|
| 1 | Oracle | 5.8759(0.0834) | 1.0454(0.0149) | 0.0245(0.0086) |
| | CMR | 5.8761(0.0673) | 1.0459(0.0123) | 0.0249(0.0071) |
| | OMR | 5.9012(0.0701) | 1.0581(0.0162) | 0.0290(0.0091) |
| 2 | Oracle | 23.464(0.3237) | 1.0441(0.0148) | 0.0926(0.0342) |
| | CMR | 23.465(0.2598) | 1.0446(0.0121) | 0.0928(0.0279) |
| | OMR | 23.580(0.2832) | 1.0573(0.0170) | 0.1115(0.0365) |
| 4 | Oracle | 93.532(0.8843) | 1.0418(0.0962) | 0.3342(0.1255) |
| | CMR | 93.542(0.9794) | 1.0421(0.0118) | 0.3346(0.1063) |
| | OMR | 94.094(1.0978) | 1.0550(0.0166) | 0.4125(0.1417) |

Table 4.3: Quantitive comparison of the statistical performance between CMR and OMR with the noise matrices generated using $\mathbf{D}_H$. The results are averaged over 200 simulations with the standard errors in parentheses. CMR and OMR achieve similar performance.

| $\sigma_{\max}$ | Method | Pre. Err. | Adj. Pre.Err | Est. Err. |
|---|---|---|---|---|
| 1 | CMR | 13.565(0.1408) | 1.0435(0.0108) | 0.0599(0.0164) |
| | OMR | 13.697(0.1554) | 1.0486(0.0142) | 0.0607(0.0128) |
| 2 | CMR | 54.171(0.5771) | 1.0418(0.0110) | 0.2252(0.0649) |
| | OMR | 54.221(0.6173) | 1.0427(0.0118) | 0.2359(0.0821) |
| 4 | CMR | 215.98(2.104) | 1.0384(0.0101) | 0.80821(0.25078) |
| | OMR | 216.19(2.391) | 1.0394(0.0114) | 0.81957(0.31806) |

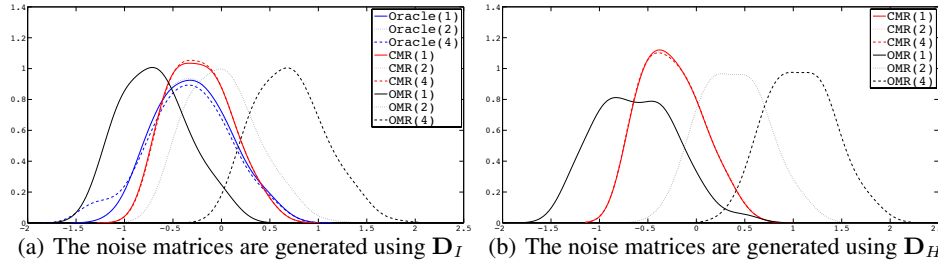

(a) The noise matrices are generated using $\mathbf{D}_I$  (b) The noise matrices are generated using $\mathbf{D}_H$

Figure 4.1: The distributions of the selected regularization parameters using the kernel density estimator. The numbers in the parentheses are $\sigma_{\max}$'s. The optimal regularization parameters of OMR are spreader with different $\sigma_{\max}$ than those of CMR and the oracle estimator.

## 5 Real Data Experiment

We apply CMR on a brain activity prediction problem which aims to build a parsimonious model to predict a person's neural activity when seeing a stimulus word. As is illustrated in Figure 5.1, for a given stimulus word, we first encode it into an intermediate semantic feature vector using some corpus statistics. We then model the brain's neural activity pattern using CMR. Creating such a predictive model not only enables us to explore new analytical tools for the fMRI data, but also helps us to gain deeper understanding on how human brain represents knowledge [16].

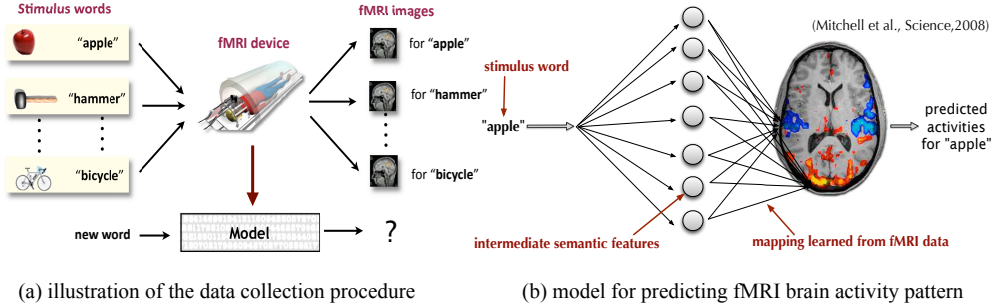

(a) illustration of the data collection procedure      (b) model for predicting fMRI brain activity pattern

Figure 5.1: An illustration of the fMRI brain activity prediction problem [16]. (a) To collect the data, a human participant sees a sequence of English words and their images. The corresponding fMRI images are recorded to represent the brain activity patterns; (b) To build a predictive model, each stimulus word is encoded into intermediate semantic features (e.g. the co-occurrence statistics of this stimulus word in a large text corpus). These intermediate features can then be used to predict the brain activity pattern.

Our experiments involves 9 participants, and Table 5.1 summarizes the prediction performance of different methods on these participants. We see that the prediction based on the features selected by CMR significantly outperforms that based on the features selected by OMR, and is as competitive as that based on the handcrafted features selected by human experts. But due to the space limit, we present the details of the real data experiment in the technical report version.

Table 5.1: Prediction accuracies of different methods (higher is better). CMR outperforms OMR for 8 out of 9 participants, and outperforms the handcrafted basis words for 6 out of 9 participants

| Method | P. 1 | P. 2 | P. 3 | P. 4 | P. 5 | P. 6 | P. 7 | P. 8 | P. 9 |
|---|---|---|---|---|---|---|---|---|---|
| CMR | 0.840 | 0.794 | 0.861 | 0.651 | 0.823 | 0.722 | 0.738 | 0.720 | 0.780 |
| OMR | 0.803 | 0.789 | 0.801 | 0.602 | 0.766 | 0.623 | 0.726 | 0.749 | 0.765 |
| Handcraft | 0.822 | 0.776 | 0.773 | 0.727 | 0.782 | 0.865 | 0.734 | 0.685 | 0.819 |

## 6 Discussions

A related method is the square-root sparse multivariate regression [8]. They solve the convex program with the Frobenius loss function and $L_{1,p}$ regularization function

$$\widehat{\mathbf{B}} = \underset{\mathbf{B}}{\operatorname{argmin}} ||\mathbf{Y} - \mathbf{XB}||_{\mathrm{F}} + \lambda||\mathbf{B}||_{1,p}. \qquad (6.1)$$

The Frobenius loss function in (6.1) makes the regularization parameter selection independent of $\sigma_{\max}$, but it does not calibrate different regression tasks. Note that we can rewrite (6.1) as

$$(\widehat{\mathbf{B}}, \widehat{\sigma}) = \underset{\mathbf{B},\sigma}{\operatorname{argmin}} \ \frac{1}{\sqrt{nm}\sigma}||\mathbf{Y} - \mathbf{XB}||_{\mathrm{F}}^2 + \lambda||\mathbf{B}||_{1,p} \ \text{ s. t. } \ \sigma = \frac{1}{\sqrt{nm}}||\mathbf{Y} - \mathbf{XB}||_{\mathrm{F}}. \qquad (6.2)$$

Since $\sigma$ in (6.2) is not specific to any individual task, it cannot calibrate the regularization. Thus it is fundamentally different from CMR.

## Footnotes

[1]The rate of convergence is optimal when $p = 2$, i.e., the regularization function is $||\mathbf{B}||_{1,p}$

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
