[Supplementary Material]

# A  Technical Proofs Related to Computational Algorithm

## A.1  Proof of Theorem 2.2

*Proof.* We consider the following decomposition

$$||\mathbf{Y} - \mathbf{XB}^{(t)}||_{2,1} + \lambda||\mathbf{B}^{(t)}||_{1,p} - ||\mathbf{Y} - \mathbf{X}\widehat{\mathbf{B}}||_{2,1} - \lambda||\widehat{\mathbf{B}}||_{1,p} = ||\mathbf{Y} - \mathbf{XB}^{(t)}||_{2,1} + \lambda||\mathbf{B}^{(t)}||_{1,p}$$
$$- ||\mathbf{Y} - \mathbf{X}\widetilde{\mathbf{B}}||_\mu - \lambda||\widetilde{\mathbf{B}}||_{1,p} + ||\mathbf{Y} - \mathbf{X}\widetilde{\mathbf{B}}||_\mu + \lambda||\widetilde{\mathbf{B}}||_{1,p} - ||\mathbf{Y} - \mathbf{X}\widehat{\mathbf{B}}||_{2,1} - \lambda||\widehat{\mathbf{B}}||_{1,p}. \quad \text{(A.1)}$$

By (2.6), we have

$$||\mathbf{Y} - \mathbf{XB}^{(t)}||_{2,1} \leq \frac{m\mu}{2} + ||\mathbf{Y} - \mathbf{XB}^{(t)}||_\mu \quad \text{and} \quad ||\mathbf{Y} - \mathbf{X}\widehat{\mathbf{B}}||_{2,1} \geq ||\mathbf{Y} - \mathbf{X}\widehat{\mathbf{B}}||_\mu. \quad \text{(A.2)}$$

Combining (A.1) and(A.2), we have

$$||\mathbf{Y} - \mathbf{XB}^{(t)}||_{2,1} + \lambda||\mathbf{B}^{(t)}||_{1,p} - ||\mathbf{Y} - \mathbf{X}\widehat{\mathbf{B}}||_{2,1} - \lambda||\widehat{\mathbf{B}}||_{1,p}$$
$$\leq \frac{m\mu}{2} + ||\mathbf{Y} - \mathbf{XB}^{(t)}||_\mu + \lambda||\mathbf{B}^{(t)}||_{1,p} - ||\mathbf{Y} - \mathbf{X}\widetilde{\mathbf{B}}||_\mu - \lambda||\widetilde{\mathbf{B}}||_{1,p}$$
$$+ ||\mathbf{Y} - \mathbf{X}\widetilde{\mathbf{B}}||_\mu + \lambda||\widetilde{\mathbf{B}}||_{1,p} - ||\mathbf{Y} - \mathbf{X}\widehat{\mathbf{B}}||_\mu - \lambda||\widehat{\mathbf{B}}||_{1,p}. \quad \text{(A.3)}$$

Since $\widetilde{\mathbf{B}}$ is the minimizer of (2.8), we have

$$||\mathbf{Y} - \mathbf{X}\widetilde{\mathbf{B}}||_\mu + \lambda||\widetilde{\mathbf{B}}||_{1,p} \leq ||\mathbf{Y} - \mathbf{X}\widehat{\mathbf{B}}||_\mu + \lambda||\widehat{\mathbf{B}}||_{1,p}. \quad \text{(A.4)}$$

By Theorem 5.1 in [4], we have

$$||\mathbf{Y} - \mathbf{XB}^{(t)}||_\mu + \lambda||\mathbf{B}^{(t)}||_{1,p} - ||\mathbf{Y} - \mathbf{X}\widetilde{\mathbf{B}}||_\mu - \lambda||\widetilde{\mathbf{B}}||_{1,p} \leq \frac{2\gamma||\mathbf{B}^{(0)} - \widetilde{\mathbf{B}}||_F^2}{\mu(t+1)^2}. \quad \text{(A.5)}$$

Note that (A.5) implies that given a pre-specified accuracy $\epsilon$, after

$$t = ||\mathbf{B}^{(0)} - \widetilde{\mathbf{B}}||_F \sqrt{2\gamma}/\sqrt{\mu\epsilon} - 1 = \mathcal{O}(1/\sqrt{\mu\epsilon}) \quad \text{(A.6)}$$

iterations, we have $||\mathbf{Y} - \mathbf{XB}^{(t)}||_\mu + \lambda||\mathbf{B}^{(t)}||_{1,p} - ||\mathbf{Y} - \mathbf{X}\widetilde{\mathbf{B}}||_\mu - \lambda||\widetilde{\mathbf{B}}||_{1,p} \leq \epsilon$. By combining (A.3), (A.4) and (A.5), we have

$$||\mathbf{Y} - \mathbf{XB}^{(t)}||_{2,1} + \lambda||\mathbf{B}^{(t)}||_{1,p} - ||\mathbf{Y} - \mathbf{X}\widehat{\mathbf{B}}||_{2,1} - \lambda||\widehat{\mathbf{B}}||_{1,p} \leq \frac{m\mu}{2} + \frac{2\gamma||\mathbf{B}^{(0)} - \widetilde{\mathbf{B}}||_F^2}{\mu(t+1)^2}. \quad \text{(A.7)}$$

Since $\mu = \epsilon/2m$, to make L.H.S. of (A.7) no smaller than $\epsilon$, we need

$$\frac{2m\gamma||\mathbf{B}^{(0)} - \widetilde{\mathbf{B}}||_F^2}{\epsilon(t+1)^2} \leq \frac{\epsilon}{2}.$$

By solving the inequality above, we obtain

$$t \geq \frac{2\sqrt{m\gamma}||\mathbf{B}^{(0)} - \widetilde{\mathbf{B}}||_F}{\epsilon} - 1,$$

which completes the proof. □

## A.2  ADMM Solver for CMR

We give a brief derivation of the alternating direction method of multipliers (ADMM) for solving CMR. We first reparametrize (2.1) as follows,

$$(\widehat{\mathbf{B}}, \widehat{\mathbf{R}}) = \underset{\mathbf{B}, \mathbf{R}}{\mathrm{argmin}} \ ||\mathbf{R}||_{2,1} + \lambda||\mathbf{B}||_{1,p} \quad \text{subject to:} \ \mathbf{Y} - \mathbf{XB} = \mathbf{R}.$$

Then for $t = 1, 2, ...$, ADMM adopts the iterative scheme

$$\mathbf{B}^{(t)} = \underset{\mathbf{B}}{\mathrm{argmin}} \ \frac{\lambda}{\rho}||\mathbf{B}||_{1,p} + \frac{1}{2}||\mathbf{U}^{(t-1)}/\rho + \mathbf{Y} - \mathbf{R}^{(t-1)} - \mathbf{XB}||_F^2, \quad \text{(A.8)}$$

$$\mathbf{R}^{(t)} = \underset{\mathbf{R}}{\mathrm{argmin}} \ \frac{1}{\rho}||\mathbf{R}||_{2,1} + \frac{1}{2}||\mathbf{U}^{(t-1)}/\rho + \mathbf{Y} - \mathbf{R} - \mathbf{XB}^{(t)}||_F^2, \quad \text{(A.9)}$$

$$\mathbf{U}^{(t)} = \mathbf{U}^{(t-1)} + \rho\left(\mathbf{Y} - \mathbf{R}^{(t)} - \mathbf{XB}^{(t)}\right). \quad \text{(A.10)}$$

where $\rho$ is a penalty parameter and $\mathbf{U}$ is the Lagrange multiplier matrix. The algorithm stops when

$$\max\left\{||\mathbf{B}^{(t)} - \mathbf{B}^{(t-1)}||_{\mathrm{F}}, \ ||\mathbf{R}^{(t)} - \mathbf{R}^{(t-1)}||_{\mathrm{F}}, \ ||\mathbf{U}^{(t)} - \mathbf{U}^{(t-1)}||_{\mathrm{F}}\right\} \leq \varepsilon,$$

where $\varepsilon$ is the stopping precision. By adopting the group soft thresholding procedure, (A.9) has a closed form solution as follows,

$$\mathbf{R}_{*k}^{(t)} = \widetilde{\mathbf{R}}_{*k}^{(t)} \cdot \max\{1 - 1/(\rho||\widetilde{\mathbf{R}}_{*k}||_2), \ 0\},$$

where $\widetilde{\mathbf{R}} = \mathbf{U}^{(t-1)}/\rho + \mathbf{Y} - \mathbf{X}\mathbf{B}^{(t)}$. There are multiple choices to solve (A.8). Let $\widetilde{\mathbf{Y}} = \mathbf{U}^{(t-1)}/\rho + \mathbf{Y} - \mathbf{R}^{(t-1)}$, then (A.8) can be rewritten as

$$\mathbf{B}^{(t)} = \underset{\mathbf{B}}{\operatorname{argmin}} \ \frac{1}{2}||\widetilde{\mathbf{Y}} - \mathbf{X}\mathbf{B}||_{\mathrm{F}}^2 + \frac{\lambda}{\rho}||\mathbf{B}||_{1,p}. \tag{A.11}$$

(A.11) is equivalent to (1.1) in the sense of optimization, therefore it can also be solved by existing OMR solvers. While a more efficient alternative is to approximately solve (A.8) using a linearization step at $\mathbf{B} = \mathbf{B}^{(t-1)}$ as follows,

$$\mathbf{B}^{(t)} = \underset{\mathbf{B}}{\operatorname{argmin}} \ \frac{\lambda}{\rho}||\mathbf{B}||_{1,p} + \frac{1}{2\eta}||\mathbf{B} - \widetilde{\mathbf{B}}||_{\mathrm{F}}^2, \tag{A.12}$$

where $\widetilde{\mathbf{B}} = \mathbf{B}^{t-1} - \eta(\mathbf{X}^T\mathbf{X}\mathbf{B}^{t-1} - \widetilde{\mathbf{Y}}^T\mathbf{X})$ and $\eta$ is a positive constant such that

$$\frac{1}{2}||\widetilde{\mathbf{Y}} - \mathbf{X}\mathbf{B}^{(t)}||_{\mathrm{F}}^2 \leq \frac{1}{2}||\widetilde{\mathbf{Y}} - \mathbf{X}\mathbf{B}^{(t-1)}||_{\mathrm{F}}^2 + \langle\mathbf{X}^T\mathbf{X}\mathbf{B}^{t-1} - \widetilde{\mathbf{Y}}^T\mathbf{X}, \mathbf{B}^{(t)} - \mathbf{B}^{(t-1)}\rangle + \frac{1}{2\eta}||\mathbf{B}^{(t)} - \mathbf{B}^{(t-1)}||_{\mathrm{F}}^2.$$

A conservative choice is $\eta = 1/||\mathbf{X}||_2^2$, and we can improve the empirical performance by the backtracking line search as is shown in Section 3. When $p = 2$, we can obtain the closed form solution to (A.12) by the group soft thresholding procedure

$$\mathbf{B}_{j*}^{(t)} = \widetilde{\mathbf{B}}_{j*} \cdot \max\{1 - \eta\lambda/(\rho||\widetilde{\mathbf{B}}_{j*}||_2), \ 0\}.$$

More details about other choices of $p$ can be found in [11, 12].

# B  Technical Proofs Related to Statistical Properties

Note that the following two relations are frequently used in our analysis,

$$\mathbf{Y} - \mathbf{X}\mathbf{B}^0 = \mathbf{X}\mathbf{B}^0 + \mathbf{Z} - \mathbf{X}\mathbf{B}^0 = \mathbf{Z} \ \text{ and } \ \mathbf{Y} - \mathbf{X}\widehat{\mathbf{B}} = \mathbf{X}\mathbf{B}^0 + \mathbf{Z} - \mathbf{X}\widehat{\mathbf{B}} = \mathbf{Z} - \mathbf{X}\widehat{\boldsymbol{\Delta}}.$$

We then present the proof of the main theorem.

## B.1  Proof of Lemma 3.1

*Proof.* By triangle inequality, we have

$$||\widehat{\mathbf{B}}||_{1,p} = ||\mathbf{B}^0 + \widehat{\boldsymbol{\Delta}}||_{1,p} = ||\mathbf{B}_{\mathcal{S}}^0 + \mathbf{B}_{\mathcal{N}}^0 + \widehat{\boldsymbol{\Delta}}_{\mathcal{S}} + \widehat{\boldsymbol{\Delta}}_{\mathcal{N}}||_{1,p}$$
$$\geq ||\mathbf{B}_{\mathcal{S}}^0 + \widehat{\boldsymbol{\Delta}}_{\mathcal{N}}||_{1,p} - ||\mathbf{B}_{\mathcal{N}}^0 + \widehat{\boldsymbol{\Delta}}_{\mathcal{S}}||_{1,p} \geq ||\mathbf{B}_{\mathcal{S}}^0||_{1,p} + ||\widehat{\boldsymbol{\Delta}}_{\mathcal{N}}||_{1,p} - ||\mathbf{B}_{\mathcal{N}}^0||_{1,p} - ||\widehat{\boldsymbol{\Delta}}_{\mathcal{S}}||_{1,p}. \tag{B.1}$$

Since $\mathbf{B}^0 \in \mathcal{S}$, we have $||\mathbf{B}_{\mathcal{N}}^0||_{1,p} = \mathbf{0}$, and $||\mathbf{B}^0||_{1,p} = ||\mathbf{B}_{\mathcal{S}}^0||_{1,p} + ||\mathbf{B}_{\mathcal{N}}^0||_{1,p} = ||\mathbf{B}_{\mathcal{S}}^0||_{1,p}$. By rearranging (B.1), we obtain

$$||\mathbf{B}^0||_{1,p} - ||\mathbf{B}^0 + \widehat{\boldsymbol{\Delta}}||_{1,p} \leq ||\widehat{\boldsymbol{\Delta}}_{\mathcal{S}}||_{1,p} - ||\widehat{\boldsymbol{\Delta}}_{\mathcal{N}}||_{1,p}. \tag{B.2}$$

Since $\widehat{\mathbf{B}}$ is the minimizer to (2.1), by (B.2), we further have

$$||\mathbf{X}\widehat{\boldsymbol{\Delta}} - \mathbf{Z}||_{2,1} - ||\mathbf{Z}||_{2,1} \leq \lambda(||\mathbf{B}^0||_{1,p} - ||\mathbf{B}^0 + \widehat{\boldsymbol{\Delta}})||_{1,p} \leq \lambda(||\widehat{\boldsymbol{\Delta}}_{\mathcal{S}}||_{1,p} - ||\widehat{\boldsymbol{\Delta}}_{\mathcal{N}}||_{1,p}). \tag{B.3}$$

Due to the convexity of $||\cdot||_{2,1}$, we know

$$||\mathbf{X}\widehat{\boldsymbol{\Delta}} - \mathbf{Z}||_{2,1} - ||\mathbf{Z}||_{2,1} \geq \langle\mathbf{G}^0, \widehat{\boldsymbol{\Delta}}\rangle \geq -|\langle\mathbf{G}^0, \widehat{\boldsymbol{\Delta}}\rangle|. \tag{B.4}$$

By the Cauchy-Schwarz inequality, we obtain

$$|\langle\mathbf{G}^0, \widehat{\boldsymbol{\Delta}}\rangle| \leq ||\mathbf{G}^0||_{\infty,q}||\widehat{\boldsymbol{\Delta}}||_{1,p} \leq \frac{\lambda}{c}(||\widehat{\boldsymbol{\Delta}}_{\mathcal{S}}||_{1,p} + ||\widehat{\boldsymbol{\Delta}}_{\mathcal{N}}||_{1,p}), \tag{B.5}$$

where the last inequality comes from the assumption $\lambda \geq c||\mathbf{G}^0||_{\infty,q}$. By combining (B.3), (B.4), and (B.5), we obtain

$$-\frac{\lambda}{c}(||\widehat{\boldsymbol{\Delta}}_{\mathcal{S}}||_{1,p} + ||\widehat{\boldsymbol{\Delta}}_{\mathcal{N}}||_{1,p}) \leq \lambda(||\widehat{\boldsymbol{\Delta}}_{\mathcal{S}}||_{1,p} - ||\widehat{\boldsymbol{\Delta}}_{\mathcal{N}}||_{1,p}). \tag{B.6}$$

By rearranging (B.6), we obtain $||\widehat{\boldsymbol{\Delta}}_{\mathcal{N}}||_{1,p} \leq (c+1)||\widehat{\boldsymbol{\Delta}}_{\mathcal{S}}||_{1,p}/(c-1)$, which completes proof. $\qquad\square$

## B.2 Proof of Theorem 3.2

*Proof.* We first assume $\lambda \geq c||\mathbf{G}^0||_{\infty,q}$. Then we have

$$||\mathbf{X}\widehat{\boldsymbol{\Delta}} - \mathbf{Z}||_{2,1} - ||\mathbf{Z}||_{2,1} = \sum_{k=1}^{m}(||\mathbf{X}\widehat{\boldsymbol{\Delta}}_{*k} - \mathbf{Z}_{*k}||_2 - ||\mathbf{Z}_{*k}||_2)$$

$$= \sum_{k=1}^{m}\frac{||\mathbf{X}\widehat{\boldsymbol{\Delta}}_{*k}||_2^2 - 2(\mathbf{X}\widehat{\boldsymbol{\Delta}}_{*k})^T\mathbf{Z}_{*k}}{||\mathbf{X}\widehat{\boldsymbol{\Delta}}_{*k} - \mathbf{Z}_{*k}||_2 + ||\mathbf{Z}_{*k}||_2} \geq \sum_{k=1}^{m}\frac{||\mathbf{X}\widehat{\boldsymbol{\Delta}}_{*k}||_2^2}{||\mathbf{X}\widehat{\boldsymbol{\Delta}}_{*k}||_2 + 2||\mathbf{Z}_{*k}||_2} - 2\sum_{k=1}^{m}\frac{|\widehat{\boldsymbol{\Delta}}_{*k}^T\mathbf{X}^T\mathbf{Z}_{*k}|}{||\mathbf{Z}_{*k}||_2}. \quad (B.7)$$

Since $\mathbf{G}_{*k}^0 = \mathbf{X}^T\mathbf{Z}_{*k}/||\mathbf{Z}_{*k}||_2$, we have

$$\sum_{k=1}^{m}\frac{|\widehat{\boldsymbol{\Delta}}_{*k}^T\mathbf{X}^T\mathbf{Z}_{*k}|}{||\mathbf{Z}_{*k}||_2} = \sum_{k=1}^{m}|\widehat{\boldsymbol{\Delta}}_{*k}^T\mathbf{G}_{*k}^0| \leq \sum_{k=1}^{m}\sum_{j=1}^{d}|\widehat{\boldsymbol{\Delta}}_{jk}\mathbf{G}_{jk}^0| \leq ||\mathbf{G}^0||_{\infty,q}||\widehat{\boldsymbol{\Delta}}||_{1,p}, \quad (B.8)$$

where the last inequality follows from the Cauchy-Schwarz inequality. Recall that in the proof of Lemma 3.1, we already have (B.3) as follows,

$$||\mathbf{X}\widehat{\boldsymbol{\Delta}} - \mathbf{Z}||_{2,1} - ||\mathbf{Z}||_{2,1} \leq \lambda(||\widehat{\boldsymbol{\Delta}}_{\mathcal{S}}||_{1,p} - ||\widehat{\boldsymbol{\Delta}}_{\mathcal{N}}||_{1,p}). \quad (B.9)$$

Therefore by combining (B.9), (B.7), and (B.8), we obtain

$$\sum_{k=1}^{m}\frac{||\mathbf{X}\widehat{\boldsymbol{\Delta}}_{*k}||_2^2}{||\mathbf{X}\widehat{\boldsymbol{\Delta}}_{*k}||_2 + 2||\mathbf{Z}_{*k}||_2} \leq \lambda(||\widehat{\boldsymbol{\Delta}}_{\mathcal{S}}||_{1,p} - ||\widehat{\boldsymbol{\Delta}}_{\mathcal{N}}||_{1,p}) + 2||\mathbf{G}^0||_{\infty,q}||\widehat{\boldsymbol{\Delta}}||_{1,p}$$

$$\leq \lambda(1 + 2/c)||\widehat{\boldsymbol{\Delta}}_{\mathcal{S}}||_{1,p} + \lambda(2/c - 1)||\widehat{\boldsymbol{\Delta}}_{\mathcal{N}}||_{1,p} \leq \frac{2\lambda}{c-1}||\widehat{\boldsymbol{\Delta}}_{\mathcal{S}}||_{1,p}, \quad (B.10)$$

where the second inequality comes from the assumption $\lambda \geq c||\mathbf{G}^0||_{\infty,q}$, and the last inequality comes from (3.3) in Lemma 3.1. Meanwhile, by triangle inequality, we also have

$$\sum_{k=1}^{m}\frac{||\mathbf{X}\widehat{\boldsymbol{\Delta}}_{*k}||_2^2}{||\mathbf{X}\widehat{\boldsymbol{\Delta}}_{*k}||_2 + 2||\mathbf{Z}_{*k}||_2} \geq \frac{\sum_{k=1}^{m}||\mathbf{X}\widehat{\boldsymbol{\Delta}}_{*k}||_2^2}{||\mathbf{X}\widehat{\boldsymbol{\Delta}}||_{2,\infty} + 2||\mathbf{Z}||_{2,\infty}} \geq \frac{||\mathbf{X}\widehat{\boldsymbol{\Delta}}||_F^2}{||\mathbf{X}\widehat{\boldsymbol{\Delta}}||_F + 2||\mathbf{Z}||_{2,\infty}}, \quad (B.11)$$

where the last inequality comes from the fact $||\mathbf{X}\widehat{\boldsymbol{\Delta}}||_{2,\infty} \leq ||\mathbf{X}\widehat{\boldsymbol{\Delta}}||_F$. Combining (B.10) and (B.11), we obtain

$$\frac{||\mathbf{X}\widehat{\boldsymbol{\Delta}}||_F^2}{||\mathbf{X}\widehat{\boldsymbol{\Delta}}||_F + 2||\mathbf{Z}||_{2,\infty}} \leq \frac{2\lambda}{c-1}||\widehat{\boldsymbol{\Delta}}_{\mathcal{S}}||_{1,p} \leq \frac{2\lambda\sqrt{s}||\widehat{\boldsymbol{\Delta}}||_F}{c-1}, \quad (B.12)$$

where the last inequality comes from the fact that $\mathcal{S}$ contains only $s$ rows with nonzero entries. By Assumption 3.1, we can rewrite (B.12) as

$$||\mathbf{X}\widehat{\boldsymbol{\Delta}}||_F^2 \leq \frac{2\lambda\sqrt{s}}{(c-1)\sqrt{n}\kappa}||\mathbf{X}\widehat{\boldsymbol{\Delta}}||_F^2 + \frac{4\lambda\sqrt{s}}{\sqrt{n}\kappa(c-1)}||\mathbf{Z}||_{2,\infty}||\mathbf{X}\widehat{\boldsymbol{\Delta}}||_F.$$

Given $2\lambda\sqrt{s} \leq (c-1)\sqrt{n}\kappa/2$, we have

$$||\mathbf{X}\widehat{\boldsymbol{\Delta}}||_F \leq \frac{8\lambda\sqrt{s}}{\sqrt{n}\kappa(c-1)}||\mathbf{Z}||_{2,\infty} \leq \frac{8\lambda\sqrt{s}\sigma_{\max}}{\sqrt{n}\kappa(c-1)}||\mathbf{W}||_{2,\infty}. \quad (B.13)$$

By Assumption 3.1 again, we obtain

$$||\widehat{\boldsymbol{\Delta}}||_F \leq \frac{8\lambda\sqrt{s}\sigma_{\max}}{n\kappa^2(c-1)}||\mathbf{W}||_{2,\infty}. \quad (B.14)$$

Now we introduce the following lemmas to deliver the concrete rates of convergence in parameter estimation.

**Lemma B.1.** *Suppose that we have all entries of a random vector* $\boldsymbol{v} = (v_1, ..., v_n)^T \in \mathbb{R}^n$ *independently generated from the standard Gaussian distribution with mean* $0$ *and variance* $1$. *For any* $c_0 \in (0, 1)$, *we have*

$$\mathbb{P}\left(\left|||\boldsymbol{v}||_2^2 - n\right| \geq c_0 n\right) \leq 2\exp\left(-\frac{nc_0^2}{8}\right).$$

The proof of Lemma B.1 is provided in [9], therefore omitted.

**Lemma B.2.** *Suppose that we have all entries of* $\mathbf{W}$ *independently generated from the standard Gaussian distribution with mean* $0$ *and variance* $1$*, then we have*

$$\mathbb{P}\left(\max_{1 \le j \le d} \frac{1}{\sqrt{n}} ||\mathbf{X}_{*j}^T \mathbf{W}||_q \le 2\left(m^{1-1/p} + \sqrt{\log d}\right)\right) \ge 1 - \frac{2}{d^2},$$

*where* $1/p + 1/q = 1$.

The proof of Lemma B.2 is provided in Appendix B.3. Now we proceed to derive the refined error bound for the joint sparsity setting.

Since we have all entries of $\mathbf{W}$ independently generated from some standard Gaussian distribution with mean $0$ and variance $1$, then by Lemma B.1, for any $c_0 \in (0, 1)$, we have

$$\mathbb{P}\left(\sqrt{(1-c_0)n} \le ||\mathbf{W}_{*k}||_2 \le \sqrt{(1+c_0)n}\right) \ge 1 - 2\exp\left(-\frac{nc_0^2}{8}\right).$$

By taking the union bound over all $k = 1, ..., m$, we have

$$\mathbb{P}\left(\sqrt{(1-c_0)n} \le \min_{1 \le k \le m} ||\mathbf{W}_{*k}||_2 \le \max_{1 \le k \le m} ||\mathbf{W}_{*k}||_2 \le \sqrt{(1+c_0)n}\right)$$
$$\ge 1 - 2m\exp\left(-\frac{nc_0^2}{8}\right). \quad \text{(B.15)}$$

Now conditioning on the event $\sqrt{(1-c_0)n} \le \min_{1 \le k \le m} ||\mathbf{W}_{*k}||_2$, we have

$$\mathcal{R}^*(\mathbf{G}^0) = \max_{1 \le j \le d} \left(\sum_{k=1}^{n} \frac{(\mathbf{W}_{*k}^T \mathbf{X}_{*j})^q}{||\mathbf{W}_{*k}||_2}\right)^{1/q} \le \frac{\max_{1 \le j \le d} ||\mathbf{W}^T \mathbf{X}_{*j}||_q}{\min_{1 \le k \le m} ||\mathbf{W}_{*k}||_2} \le \frac{||\mathbf{W}^T \mathbf{X}||_{\infty,q}}{\sqrt{(1-c_0)n}}. \quad \text{(B.16)}$$

By Lemma B.2, we have

$$\mathbb{P}\left(\frac{||\mathbf{X}^T \mathbf{W}||_{\infty,q}}{\sqrt{(1-c_0)n}} \le \frac{2m^{1-1/p}}{\sqrt{(1-c_0)}} + \frac{2\sqrt{\log d}}{\sqrt{(1-c_0)}}\right) \ge 1 - \frac{2}{d^2}. \quad \text{(B.17)}$$

Since we requires

$$2\lambda\sqrt{s} \le \delta(c-1)\phi(n)\kappa \text{ for some } \delta < 1, \quad \text{(B.18)}$$

thus if we take

$$\lambda = \frac{2c(m^{1-1/p} + \sqrt{\log d})}{\sqrt{1-c_0}},$$

we need $n$ to be large enough

$$\sqrt{n} \ge \frac{4c\sqrt{s}(m^{1-1/p} + \sqrt{\log d})}{\delta(c-1)\sqrt{1-c_0}\kappa},$$

such that (B.18) can be secured. Then by combining (B.15), (B.16), (B.17), and (B.14), we have

$$\mathbb{P}\left(\frac{1}{\sqrt{m}} ||\widehat{\mathbf{B}} - \mathbf{B}^0||_F \le \frac{8c\sqrt{(1+c_0)}\sigma_{\max}}{\kappa^2(c-1)(1-\delta)\sqrt{(1-c_0)}}\left[\sqrt{\frac{sm^{1-2/p}}{n}} + \sqrt{\frac{s\log d}{nm}}\right]\right)$$
$$\ge 1 - \frac{2}{d^2} - 2m\exp\left(-\frac{nc_0^2}{8}\right).$$

This completes the proof. $\qquad\square$

## B.3 Proof of Lemma B.2

*Proof.* We adopt the similar proof strategy in [17], and begin our proof by establishing the tail bound of $||\mathbf{W}^T \mathbf{X}_{*j}||_q/\sqrt{n}$.

***Deviation above the mean***: Given any pair of $\mathbf{W}, \widetilde{\mathbf{W}} \in \mathbb{R}^{n \times m}$, we have

$$\left|\frac{1}{\sqrt{n}} ||\mathbf{W}^T \mathbf{X}_{*j}||_q - \frac{1}{\sqrt{n}} ||\widetilde{\mathbf{W}}^T \mathbf{X}_{*j}||_q\right| \le \frac{1}{\sqrt{n}} ||(\mathbf{W} - \widetilde{\mathbf{W}})^T \mathbf{X}_{*j}||_q$$
$$= \frac{1}{\sqrt{n}} \max_{||\boldsymbol{\theta}||_p \le 1} \langle \boldsymbol{\theta}, (\mathbf{W} - \widetilde{\mathbf{W}})^T \mathbf{X}_{*j}\rangle. \quad \text{(B.19)}$$

By the Cauchy-Schwartz inequality, we have

$$\frac{1}{\sqrt{n}} \max_{||\boldsymbol{\theta}||_p \leq 1} \langle \boldsymbol{\theta}\mathbf{X}_{*j}^T, \mathbf{W} - \widetilde{\mathbf{W}} \rangle \leq \frac{||\mathbf{W} - \widetilde{\mathbf{W}}||_F}{\sqrt{n}} \max_{||\boldsymbol{\theta}||_p \leq 1} ||\boldsymbol{\theta}\mathbf{X}_{*j}^T||_F. \tag{B.20}$$

Since $\boldsymbol{\theta}\mathbf{X}_{*j}^T$ is a rank one matrix, its singular value decomposition is

$$\boldsymbol{\theta}\mathbf{X}_{*j}^T = ||\boldsymbol{\theta}||_2 ||\mathbf{X}_{*j}|| \cdot \frac{\boldsymbol{\theta}}{||\boldsymbol{\theta}||_2} \cdot \frac{\mathbf{X}_{*j}^T}{||\mathbf{X}_{*j}||_2}.$$

Consequently, we have

$$\frac{1}{n} \max_{||\boldsymbol{\theta}||_p \leq 1} ||\boldsymbol{\theta}\mathbf{X}_{*j}^T||_F = \frac{||\mathbf{X}_{*j}||_2}{n} \max_{||\boldsymbol{\theta}||_p \leq 1} ||\boldsymbol{\theta}||_2 \overset{(i)}{\leq} \frac{m^{1/2-1/p}||\mathbf{X}_{*j}||_2}{\sqrt{n}} \overset{(ii)}{\leq} 1. \tag{B.21}$$

where (i) comes from $||\boldsymbol{\theta}||_2 \leq m^{1/2-1/p}||\boldsymbol{\theta}||_p$, and (ii) comes from the column normalization condition. Combining (B.19), (B.20), and (B.21), we obtain

$$\left| \frac{1}{\sqrt{n}}||\mathbf{W}^T\mathbf{X}_{*j}||_q - \frac{1}{\sqrt{n}}||\widetilde{\mathbf{W}}^T\mathbf{X}_{*j}||_q \right| \leq ||\mathbf{W} - \widetilde{\mathbf{W}}||_F. \tag{B.22}$$

which implies that $||\mathbf{W}^T\mathbf{X}_{*j}||_q/\sqrt{n}$ is a Lipschitz continuous function of $\mathbf{W}$ with a Lipschitz constant as 1. By the Gaussian concentration of measure for Lipschitz functions [10], we have

$$\mathbb{P}\left( \frac{1}{\sqrt{n}}||\mathbf{W}^T\mathbf{X}_{*j}||_q \geq \mathbb{E}\frac{1}{\sqrt{n}}||\mathbf{W}^T\mathbf{X}_{*j}||_q + \xi \right) \leq 2\exp\left(-\frac{\xi^2}{2}\right). \tag{B.23}$$

***Upper bound of the mean***: Given any $\boldsymbol{\beta} \in \mathbb{R}^m$, we define a zero mean Gaussian random variable $J_{\boldsymbol{\beta}} = \boldsymbol{\beta}^T\mathbf{W}^T\mathbf{X}_{*j}/\sqrt{n}$, and note that we have $\frac{1}{\sqrt{n}}||\mathbf{W}^T\mathbf{X}_{*j}||_q = \max_{||\boldsymbol{\beta}||_p=1} J_{\boldsymbol{\beta}}$. Thus given any two vectors $||\boldsymbol{\beta}||_p \leq 1$ and $||\boldsymbol{\beta}'||_p \leq 1$, we have

$$\mathbb{E}(J_{\boldsymbol{\beta}} - J_{\boldsymbol{\beta}'})^2 = \frac{1}{n}||\mathbf{X}_{*j}||_2^2 ||\boldsymbol{\beta} - \boldsymbol{\beta}'||_2^2 \leq ||\boldsymbol{\beta} - \boldsymbol{\beta}'||_2^2,$$

where the last inequality comes from the column normalization condition and $m^{1-1/p} \geq 1$.

Then we define another Gaussian random variable $K_{\boldsymbol{\beta}} = \boldsymbol{\beta}^T\boldsymbol{\omega}$, where $\boldsymbol{\omega} = (\omega_1, ..., \omega_m)^T \sim N(\mathbf{0}, \mathbf{I}_m)$ is standard Gaussian. By construction, for any pair $\boldsymbol{\beta}, \boldsymbol{\beta}' \in \mathbb{R}^m$, we have

$$\mathbb{E}[(K_{\boldsymbol{\beta}} - K_{\boldsymbol{\beta}'})^2] = ||\boldsymbol{\beta} - \boldsymbol{\beta}'||_2^2 \geq \mathbb{E}(J_{\boldsymbol{\beta}} - J_{\boldsymbol{\beta}'})^2.$$

Thus by the Sudakov-Fernique comparison principle [10], we have

$$\mathbb{E}\frac{1}{\sqrt{n}}||\mathbf{W}^T\mathbf{X}_{*j}||_q = \mathbb{E}\max_{||\boldsymbol{\beta}||_p=1} J_{\boldsymbol{\beta}} \leq \mathbb{E}\max_{||\boldsymbol{\beta}||_p=1} K_{\boldsymbol{\beta}}.$$

By definition of $K_{\boldsymbol{\beta}}$, we have

$$\mathbb{E}\max_{||\boldsymbol{\beta}||_p=1} K_{\boldsymbol{\beta}} = \mathbb{E}||\boldsymbol{\omega}||_q \leq m^{1/q}(\mathbb{E}|\boldsymbol{\omega}_1|^q)^{1/q}, \tag{B.24}$$

where the last inequality comes from Jensen's inequality and the fact that $|\boldsymbol{\omega}_1|^{1/q}$ is a concave function of $\boldsymbol{\omega}_1$ for $q \in [1, 2]$. Eventually, by Hölder inequality, we obtain

$$(\mathbb{E}|\boldsymbol{\omega}_1|^q)^{1/q} \leq \sqrt{\mathbb{E}\omega_1^2} = 1. \tag{B.25}$$

Combing (B.24) and (B.25), we obtain

$$\mathbb{E}\max_{||\boldsymbol{\beta}||_p=1} K_{\boldsymbol{\beta}} \leq m^{1-1/p} \leq 2m^{1-1/p}. \tag{B.26}$$

Then combing (B.23) and (B.26), we have

$$\mathbb{P}\left( \frac{1}{\sqrt{n}}||\mathbf{W}^T\mathbf{X}_{*j}||_q \geq 2m^{1-1/p} + \xi \right) \leq 2\exp\left(-\frac{\xi^2}{2}\right).$$

Taking the union bound over $j = 1, ..., d$ and let $\xi = 2\sqrt{\log d}$, we have

$$\mathbb{P}\left( \frac{1}{\sqrt{n}}||\mathbf{X}^T\mathbf{W}||_{\infty,q} \geq 2m^{1-1/p} + 2\sqrt{\log d} \right) \leq \frac{2}{d}.$$

This finishes the proof. □