[Reviews · NeurIPS 2014]

Submitted by Assigned_Reviewer_15

The paper proposes a new regression method, namely calibrated multivariate regression (CMR), for high dimensional data analysis. Besides proposing the CMR formulation, the paper focuses on (1) using a smoothed proximal gradient method to compute CMR’s optimal solutions; (2) analyzing CMR’ statical properties.

One key contribution of the paper lies in the introduction of this CMR formulation; its loss term can be interpreted as calibrating each regression task’s loss term with respect to its noise level. I am wondering whether there is any more intuitive interpretation behind the use of the noise level for calibration? The authors are encouraged to explain more on this point.

The results from Theorem 3.2 shows that CMR achieves the same rates of convergences as its non-calibrated counterpart OMR. Since OMR has a differentiable loss term, OMR seems to have computational advantages compared to CMR. The authors are encouraged to provide some guides on the selection between OMR and CMR.
Summary: The paper proposes a new regression method, called CMR, for high-dimensional data analysis. The papers propose to employ the smoothed proximal method to compute its solution, and also theoretically analyze its statistical properties.

Submitted by Assigned_Reviewer_19

This paper proposed a calibrated multivariate regression method for fitting high dimensional data. Then they proposed a computational method for solving the optimization problem. The main comments on the details of the paper are as follows:
1. The main improvement of the model is substituting the Frobenious norm by L_{2,1} norm. The calibration is in fact substituting the variance as standard deviation. The authors used numerical experiments to show its effects. Can the authors give some theoretical explanation or intuitive explanation why the standard deviation works better than variance?
2. For the numerical experiments, matrix D_I is added to show the different variances of the noise. However, in the setting, the range of D_I is not large. What will the results be when the range of D_I is much larger? In Table 4.1, when lambda changes in its wide range, are there the similar results?
3. Some typing errors such as: in (1.2), a lambda is lost.
Summary: The manuscript is clearly written, but lack of novelty and enough experiments.

Submitted by Assigned_Reviewer_31

This paper proposed a new method called calibrated multivariate regression (CMR) for fitting high dimensional multivariate regression models. The assumption is that the noise matrix has an uncorrelated structure, and different regression tasks have different noise variances. Instead of using the same tuning parameter lambda for all the regression tasks, CMR calibrates different tasks by solving a penalized weighted least square problem weights define in (2.3) (i.e., some kind of estimate of noise standard deviation). The paper is well-structured and clearly-written. The appendix contains a lot of technical details. The idea of CMR formulation is technically sound. The proposed computational algorithms and demonstrated statistical properties make sense at a high level, although I did not check every step of the derivation and proof given the limited expertise in this field. The empirical study on simulated and real data yielded promising results.

Line 129: Should "weighted least square program" read "weighted least square problem"?

Another straightforward approach to calibrate the parameter lambda is to run multiple regression on each response variable separately. How does this compare with the proposed CMR? It would be interesting to compare them on both simulated and real data. Sometimes simple models might yield better or more interpretable results on real data than more advanced ones.
Summary: This paper proposed a new method called calibrated multivariate regression (CMR) for fitting high dimensional multivariate regression models. The idea is technically sound and the results are promising.
Author Feedback
Author rebuttal: We thank the reviewers for their helpful comments. We will carefully address all raised concerns in the final version. Below are some clarifications:

Reviewer 1.

(1) An intuitive explanation is that if a regression task has a larger noise level, it needs more regularization to “eliminate” the random noise to yield zero entries. Theoretically, we can show that the optimal regularization parameter should be proportional to the noise level of the linear model \sigma_k. Thus, when we jointly estimate multiple linear models using the same regularization parameter, we expect the regularization parameter to be rescaled by $\sigma_k $. It eventually leads to our calibrated formulation in (2.2). We will present the above interpretation of the calibration in the final version.

(2) Though CMR adopts a nonsmooth function, but it is not necessarily more difficult than OMR in computation. As we mentioned from Line 132-133, the L2-1 loss function is only non-differentiable when the residuals of a task are exactly zero, which is unlikely to happen in practice. In the final version, we will add theoretical and empirical justification to show that the smoothing proximal gradient algorithm can attain the exact solution with an iteration complexity of O(1/\sqrt{\epsilon}). That is the same as the fast proximal gradient algorithm for solving OMR, and faster than that in Thm 2.2 of the current version. The smoothing parameter \mu is allowed to be reasonably large (e.g., scale with \sqrt{n}, and the minimum noise level of all tasks). Thus compared with OMR, CMR has no significant computational disadvantage.

Reviewer 2:

(1) If we use the variance as the weight, there will be two major problems: (a) we will lose the computational convenience. We need more sophisticated procedures and stronger assumptions to accurately estimate the variance of each task. We cannot use the same conversion rules as (2.1)-(2.3) to reformulate the estimation problem as one single convex program, because the weight (estimated variance) and the square loss function of each task will cancel each other, which makes the minimization problem invalid. (b) Even if we use the true variance, the resulting estimator will lose the tuning insensitiveness. Its optimal regularization parameter will depend on the standard deviation, which requires extra tuning efforts.

(2) In our experiment, we found that CMR is very robust to the range of D_I. If we increase the range of D_I, CMR will perform even better than OMR. We will add more explanation and experimental results on varying the range of D_I in the final version.

Reviewer 3:

We can estimate each linear model separately, but that will lose the strength of sharing the support across tasks. We will not select the same variables for all tasks, which may not be good for model interpretation. The estimation error will also be larger than CMR both theoretically and empirically. Moreover, we will need to individually select the regularization parameter for each task, which requires extra tuning efforts.

We also thank all reviewers for pointing out the typos. We will improve the writing and fix these typos in the nest iteration of the paper.